# Curcumin–Triterpene Type Hybrid as Effective Sonosensitizers for Sonodynamic Therapy in Oral Squamous Cell Carcinoma

**DOI:** 10.3390/pharmaceutics15072008

**Published:** 2023-07-23

**Authors:** Katarzyna Sowa-Kasprzak, Małgorzata Józkowiak, Dorota Olender, Anna Pawełczyk, Hanna Piotrowska-Kempisty, Lucjusz Zaprutko

**Affiliations:** 1Chair and Department of Organic Chemistry, Poznan University of Medical Sciences, Grunwaldzka 6 Str., 60-780 Poznań, Poland; dolender@ump.edu.pl (D.O.); apaw@ump.edu.pl (A.P.); zaprutko@ump.edu.pl (L.Z.); 2Department of Toxicology, Poznan University of Medical Sciences, Dojazd 30 Str., 61-131 Poznań, Poland; malgorzata.jozkowiak@gmail.com (M.J.); hpiotrow@ump.edu.pl (H.P.-K.)

**Keywords:** curcumin, oleanolic acid, sonosensitizers, sonodynamic therapy, cytotoxic activity

## Abstract

Sonodynamic therapy (SDT) is a non-invasive therapeutic modality in cancer treatment that combines low-intensity ultrasound (US) and sonosensitizers. Tumor cells are destroyed through the synergistic effects of ultrasound and a chemical sonosensitizer. This study focused on the synthesis and in vitro evaluation of the sonodynamic effect of natural curcumin, triterpene oleanolic acid, and their semi-synthetic derivatives on tongue cancer SCC-25 and hypopharyngeal FaDu cell lines. The combination of the tested compounds with sonication showed a synergistic increase in cytotoxicity. In the group of oleanolic acid derivatives, oleanoyl hydrogen succinate (**6**) showed the strongest cytotoxic effect both in the SCC-25 and FaDu cell lines. Comparing curcumin (**4**) and its pyrazole derivative (**5**), curcumin showed a better cytotoxic effect on SCC-25 cells, while curcumin pyrazole was more potent on FaDu cells. The highest sonotherapeutic activity, compared to its individual components, was demonstrated by a structural linker mode hybrid containing both curcumin pyrazole-oleanoyl hydrogen succinate units within one complex molecule (**7**). This study can be beneficial in the context of new perspectives in the search for effective sonosensitizers among derivatives of natural organic compounds.

## 1. Introduction

In recent years, scientists have been working on developing non-invasive and effective anticancer therapies with minimal side effects, as well as on overcoming drug resistance [1,2,3,4,5]. Based on the concept of photodynamic cancer therapy (PDT) in the 1970s, the idea of ultrasound-based cancer therapy emerged. The limitation of PDT therapy turned out to be the possibility of light penetration into the cells being too low, which is needed for the activation of photosensitizers [6]. In sonodynamic therapy (SDT), chemical sonosensitizers under the influence of US radiation and in the presence of water or O_2_ generate reactive oxygen species (ROS), causing oxidative stress and then the death of cancer cells as a result of the cavitation effect [7]. It is assumed that cavitation is mainly responsible for the biological effects of ultrasound on cells, including the cytotoxic effect, which is the basis for ultrasound research as a tool supporting anticancer therapy. The sonosensitizer can enhance the effects of ultrasound irradiation; therefore, it is an important element of sonodynamic therapy. Based on the known mechanisms of SDT-induced cell apoptosis or cell death, sonosensitizers have achieved rapid development in recent years [8]. SDT has demonstrated great efficiency in a wide variety of cancer cell lines, including glioblastoma [9], lung [10], breast [11], and leukemia [12]. The biological effect of SDT on oral squamous cell carcinoma (OSCC) has also been investigated [13]. As with PDT, SDT has been shown to induce apoptosis [14] and autophagy [15]; the exact mode of cell death is dependent on the sonosensitizer, US exposure parameters, and target type.

Organic sonosensitizers, mostly derived from photosensitizers, have many advantages: they have a specific molecular structure, their syntheses are reproducible, they generate large amounts of ROS under the influence of US irradiation, which can reduce thermal and mechanical damage caused by US, and they are easily degraded and excreted from the body, showing good biocompatibility and biodegradability [16,17,18,19]. Poor solubility in water, high phototoxicity, which can lead to skin lesions, and low tumor targeting ability limit the therapeutic effect of most organic sonosensitizers. Therefore, the search for new sonosensitizers with high sonotoxicity, low phototoxicity, and better water solubility and biocompatibility, as well as strong tumor targeting capacity, is highly important in SDT therapy [7]. A variety of sensitizers have been used in SDT studies, including porphyrins [20,21], 5-aminolevulinic acid (5-ALA) [22], phthalocyanines [23], xanthenes [24], indocyanines [25], quinolone antibiotics [26,27] or natural products [28,29]. Particularly high hopes are attached to the last mentioned group of natural sonosensitizers. Some natural products have been reported to produce sonosensitive activity under US irradiation, e.g., curcumin, emodin, hypericin, hydroxysafflor yellow A, and artemisinin [30]. Interesting examples of compounds with potential use in SDT therapy are shown in Figure 1. Hydroxysafflor yellow A is a natural compound from the chalcone class, structurally similar to curcumin, which is also an effective sonosensitizer [30]. In addition, porphyrin derivatives, including heterocyclic derivatives with a pyrazole system [31], show great potential as sensitizing compounds.

The aim of the study was to synthesize natural origin compounds from the group of curcumin and oleanolic acid (OA), screen the cytotoxicity of compounds, and compare their effects after US treatment. In our study, we compared the cytotoxic effects of synthesized simple derivatives of curcumin and oleanolic acid and their hybrid structural combination in mono- and multitherapy with US in tongue cancer SCC-25 and hypopharyngeal FaDu cell lines. So far, in the scientific literature, we have not found any effect of sonodynamic therapy of this kind of compounds in the human oral squamous cell carcinoma (OSCC) model. The data from our study can be very useful in the context of the search for effective sonosensitizers among derivatives of natural compounds.

## 2. Materials and Methods

### 2.1. Equipment and General Procedures

Curcumin (abcr GmbH, Karlsruhe, Germany), oleanolic acid (Bio-Tech, Beijing, China, source: *Olea europea*), N,N′-dicycloheksylocarbodiimide (DCC), 4-(dimethylamino)pyridine (DMAP), and solvents (chloroform, dioxane, hexane, and ethyl acetate) were commercial reagents (Sigma-Aldrich, St. Louis, MO, USA). All the compounds used for the synthesis were of a purity higher than 98%. Solvents were used without further purification. The purity of the obtained compounds was evaluated based on spectral data (NMR and EI/ESI-MS).

*3β-Hydroxyolean-12-en-28-oic acid methyl ester, Oleanolic acid methyl ester* (**1a**)

Oleanolic acid methyl ester was synthesized from oleanolic acid according to the protocol in the literature [32]. Yield: 96%; m.p. and spectral data agreed with the literature [32].

*3-Oxoolean-12-en-28-oic acid (**1b**), 3-Oxoolean-12-en-28-oic acid methyl ester* (**1c**)

Oleanolic acid (**1**, 4.57 g, 0.01 mol) or oleanolic acid methyl ester (**1a**, 4.71 g, 0.01 mol) was dissolved in acetone (140 mL), and then Jones’ reagent was added dropwise. The resulting yellow-green mixture was stirred for 30 min, then isopropyl alcohol was added dropwise. The resulting green suspension was filtered off; the filtrate was poured into a 5-fold volume of water. The white precipitate obtained was filtered off, washed with water, dried, and crystallized. Yield: 94% (**1b**); 90% (**1c**) m.p. and spectral data agreed with the literature [33].

*3-Hydroxyiminoolean-12-en-28-oic acid, Oleanolic acid oxime* (**2**)

Hydroxylamine hydrochloride (3.47 g, 0.05 mol) and sodium acetate (6.56 g, 0.08 mol) were added to a saturated, hot solution of **1b** (4.55 g, 0.01 mol) in ethanol (100 mL), and the resulting mixture was refluxed for 30 min, cooled, and poured into a 5-fold volume of water. The formed white precipitate was filtered off, washed with water, dried, and crystallized. Yield: 89%; m.p. and spectral data agreed with the literature [34].

*3-Hydroxyiminoolean-12-en-28-oic acid methyl ester, Oleanolic acid oxime methyl ester* (**3**)

Hydroxylamine hydrochloride (3.47 g, 0.05 mol) and sodium acetate (6.56 g, 0.08 mol) were added to a saturated, hot solution of **1c** (4.69 g, 0.01 mol) in ethanol (100 mL), and the resulting mixture was refluxed for 30 min, cooled, and poured into a 5-fold volume of water. The formed white precipitate was filtered off, washed with water, dried, and crystallized. Yield: 93%; m.p. and spectral data agreed with the literature [35].

*3,5-Bis[β-(4-hydroxy-3-methoxyphenyl)-ethenyl]-1H-pyrazole, Curcumin pyrazole* (**5**)

A mixture of curcumin (**4**, 0.37 g, 0.001 mol), hydrazine dihydrochloride (0.10 g, 0.001 mol), and ethanol (10 mL) was refluxed for 48 h. The progress of the reaction was checked by thin layer chromatography (TLC). After complete conversion, the mixture was cooled to room temperature, and the precipitate formed was filtered under vacuum and air-dried. Then, the crude product was subjected to column chromatography on silica gel using a chloroform/methanol (9:1, *v*:*v*) mixture as eluent. The pure product was obtained as a red solid. Yield: 79%; m.p. and spectral data were in good agreement with the reference [36].

*4-{[(3β)-28-hydroksy-28-oxoolean-12-en-3-yl]oxy}-4- oxobutanoic acid, Oleanoyl hydrogen succinate* (**6**)

Oleanolic acid (**1**, 0.94 g, 0.002 mol) and 4-(dimethylamino)pyridine (2.44 g, 0.02 mol) were added to a solution of succinic anhydride (2.0 g, 0.02 mol) in dry pyridine (10 mL). Then, the mixture was refluxed for 8 h. The reaction mixture was left overnight and poured into 100 mL of water. The precipitate formed was filtered under reduced pressure, washed with 5% HCl, air-dried, and finally crystallized from ethanol. Yield: 90%; m.p. and spectral data were in good agreement with the reference [37].

*Curcumin pyrazole oleanoyl hydrogen succinate* (**7**)

To the solution of curcumin pyrazole (**5**, 0.18 g, 0.0005 mol) and N,N′-dicycloheksylocarbodiimide (0.21 g, 0.0008 mol) in dioxane, (20 mL) compound **6** (0.33 g, 0.0006 mol) and 4-(dimethylamino)pyridine (0.05 g, 0.0004 mol) were added. The reaction mixture was stirred at room temperature for 24 h. The progress of the reaction was checked by thin layer chromatography (TLC). After complete conversion, hexane (8 mL) was added, and the dicyclohexylurea formed was filtered under vacuum. The obtained organic phase was washed sequentially with 5% HCl, 5% NaHCO_3_, and water. Then, the mixture was dried over anhydrous MgSO_4_ and finally evaporated. The crude product was subjected to column chromatography on silica gel using a chloroform/methanol (20:1, *v*:*v*) mixture as eluent. Yield: 61%; m.p. and spectral data were in good agreement with the reference [38].

### 2.2. Cell Culture and Viability Cells

SCC-25 and FaDu human *squamous cell carcinoma* cell lines were commercial products from the American Type Culture Collection (ATCC). Optimal cell culture conditions in an incubator were used (temperature: 37 °C, humidity: 95%, carbon dioxide content: 5%). SCC-25 cell line was cultured in phenol red-free DMEM: F-12 medium complemented with 10% FBS, 2.5 mM L-glutamine, penicillin (100 U/mL), and streptomycin (0.1 mg/mL). FaDu cells were maintained in phenol red-free Eagle’s Minimum Essential Medium (EMEM) supplemented with 10% fetal bovine serum (FBS), 2.5 mM L-glutamine, penicillin (100 U/mL), and streptomycin (0.1 mg/mL). For experiments, confluent stock cultures were harvested using the trypsin-EDTA solution and seeded in 35 mm diameter Petri plates at a density of 1 × 10^6^. They were allowed to attach overnight, and the compounds tested were then added at a concentration of 40 µM. Control cells were maintained under the same conditions with 0.1% DMSO. After 8 h incubation, both control and tested cells were exposed to US for 3 min using a 35 mm transducer with a resonance frequency of 3 MHz and ultrasonic intensity of 0.8 W/cm^2^. To assess the viability of cells, the MTT test was performed after 6 h of sonotherapy. The suspensions were removed and replaced with MTT/EMEM or MTT/DMEM: F-12 mixture (1:8). The resulting formazan crystals were dissolved by adding 1 mL of DSMO. Absorbance was measured using an Elx-800 plate reader (BioTek, Heilbronn, Germany) at 570 nm (reference wavelength 650 nm).

## 3. Results and Discussion

There are many literature reports confirming that both curcumin and oleanolic acid can be expected to have sonotherapeutic potential. Curcumin is the most important component of the rhizomes of *Curcuma longa*. Beyond its wide range of cellular properties including antiproliferative [39], pro-apoptotic, anticancer [40], antioxidant, anti-inflammatory [41], and antibacterial activities [42], more and more pieces of evidence reveal that visible light and ultrasounds can activate curcumin to have significantly enhanced toxicity in cancer cells, which demonstrates that curcumin is a herbal photo- and sonosensitizer [8]. Studies on antitumor therapy have shown that curcumin-mediated SDT can increase the generation of ROS in nasopharyngeal carcinoma CNE2 cells [43].

Naturally occurring triterpenes are an important class of compounds with interesting biological properties. The most well-known compound produced by medicinal plants and herbs is oleanolic acid (OA) [44]. Some pharmacological activities such as antioxidant, antitumor, anti-inflammatory, antidiabetic, and antimicrobial effects have been attributed to OA in different models of diseases [45,46,47]. Due to its hepatoprotective effect, OA has been used in China as a medicine for the liver for over 20 years [48]. The search for new synthetic analogs of oleanolic acid with desired biological activities is important in the process of discovering new drugs [49]. Oleanolic acid combined with the photosensitizer chlorin e6 has been used through self-assembly technology to create a carrier-free nanosensitizer for combination chemotherapy and SPDT for cancer treatment. Under the influence of light and US irradiation, this combination showed a synergistic inhibitory effect in PC9 and 4T1 cells with a significant decrease in IC50 values [50].

Accordingly, it is expected that curcumin and/or oleanolic acid core structures may be powerful sonosensitizers. For this purpose, the sonosensitizing potential of oleanolic acid itself (**1**) and its most popular semi-synthetic derivatives, such as oleanolic acid oxime (**2**), oleanolic acid oxime methyl ester (**3**), and oleanoyl hydrogen succinate (**6**), was investigated. In parallel, similar studies were conducted for naturally occurring curcumin (**4**) and its heterocyclic pyrazole derivative (**5**) with a blocked 1,3-diketone moiety.

Moreover, it was also decided to test the sonotherapeutic properties of their mutual covalent structural combinations. The resulting curcumin–triterpene hybrid (**7**) combines the structural features of two different natural bioactive molecules, which creates the possibility of specific pharmacological properties (Figure 2).

The idea of obtaining compound **7** is consistent with the currently used hybrid strategy in the design of innovative therapeutics. In order to obtain the above hybrid **7**, the linker mode approach was used [51]. As a linking molecule, a dicarboxylic succinic acid unit was used, which can be introduced into a triterpene molecule in a relatively simple way, and then the resulting triterpene hydrogen succinate (**6**) is reacted with a heterocyclic curcumin derivative (**5**).

An additional justification for the proposed hybrid strategy for compound **7** is already known triterpene hybrid structures [52], for example, based on the covalent connection of oleanolic acid oxime and aspirin [53]. The obtained compound shows a new enhanced analgesic and anti-inflammatory biological effect, however, according to a new mechanism different from the mechanisms of individual components. In the case of curcumin, an example of an effective hybrid strategy is the thalidomide–curcumin combination, which shows strong anticancer activity [54]. The general term *molecular consortia* has been proposed for such novel multifunctional compounds [51].

### 3.1. Chemistry

So far, many synthetic oleanolic acid (OA) derivatives with more favorable anti-inflammatory and cytoprotective properties than the original OA have been obtained [55]. Biological tests also showed that the compounds with the hydroxyimino group (oximes) in the triterpene molecule belong to the class of the most active species [56]. Oleanolic acid derivatives were obtained using earlier published methods [32,33,34,35,37]. The secondary hydroxyl group at the C-3 position of oleanolic acid (**1**) was reacted by oxidation with chromic anhydride in sulfuric acid (Jones reagent) to give 3-oxo-oleanolic derivatives (**1b**, **1c**). 3-Oxoleanolic derivatives (**1b**, **1c**) were obtained by reacting the hydroxyl group of oleanolic acid (**1**) with chromic anhydride in the presence of sulfuric acid (Jones reagent). Subsequently, the carbonyl derivatives were reacted with hydroxylamine hydrochloride in the presence of sodium acetate and converted into the corresponding oximes (**2**, **3**). In order to obtain the ester derivative (**1a**), the carboxyl group of oleanolic acid (**1**) at the C-28 position was converted into the methyl ester using dimethyl sulfate in an ethanolic solution of sodium hydroxide as a methylating agent [32]. The presence of the C-28 carboxyl group in the triterpene molecule increases the polarity of such a derivative but, at the same time, reduces the solubility in organic solvents and those used in cytotoxic studies. The ester derivatives are better soluble in many organic solvents. Oleanoyl hydrogen succinate (**6**) was prepared in the reaction of oleanolic acid and succinic anhydride. The process was catalyzed by the nucleophilic 4-(dimethylamino)pyridine (DMAP). The introduction of the succinic linker allowed for a change in the functionality of the triterpene unit, and the obtained derivatives with a carboxyl group were successfully used for the synthesis of hybrid derivatives with potential bioactivity [37]. Figure 1 presents an overview of the synthesis.

The pharmacological activity of curcumin is improved by replacing the dicarbonyl system with isosteric isoxazole or pyrazole. The SAR analysis confirmed that the introduction of a hydrazine molecule into the keto-enol moiety of curcumin improves antitumor activity [57]. The heterocyclic pyrazole derivative of curcumin was obtained based on the method described in the literature [58]. Curcumin pyrazole (**5**) was prepared by heterocyclization of a diketone system with hydrazine dihydrochloride (Figure 2).

In previous work [38], selected monoester hybrid-type derivatives based on curcumin or its heterocyclic analogs (pyrazole or isoxazole) and oleanolic acid with the succinic acid moiety were obtained. In this case, the synthesis of the hybrid derivative **7** was carried out by reacting the phenolic group of the curcumin pyrazole (**5**) with the free carboxyl group of the 4-carbon linker oleanoyl hydrogen succinate (**6**) in the presence of dicyclohexylcarbodiimide (DCC) and 4-(dimethylamino)pyridine (DMAP) (Figure 3).

^1^H-, ^13^C-NMR, and EI/ESI-MS spectral data were used to confirm the identity and structure of the obtained compounds. The recorded signals correspond well with the structure of curcumin and oleanolic acid.

General information on the conducted chemical experiments, as well as a description of the chemical structure of the synthesized compounds, can be found in our previous publication [35,38].

### 3.2. The Cytotoxic Effects of Compounds Tested Combined with US on SCC-25 and FaDu Cell Lines

*Squamous cell carcinoma (SCC*) is one of the most common cancers of the neck and head [59] and usually occurs in the oral cavity and pharyngeal regions. Sonosensitizers, as essential elements of effective sonodynamic therapy, have been used in various cancers, including oral SCC [13]. SDT has been suggested to be used as a supportive treatment for small tumors located in the head and neck region [60]]. Furthermore, several studies have shown the positive effects of sonotherapy in cell lines derived from these organs [13,61,62,63]. Hence, SCC-25 and FaDu cell lines were chosen for our study. An important issue of SDT is the selection of appropriate concentrations of the compounds tested. Based on the available literature data [12,61,64] and our preliminary studies, we chose the lowest concentrations of curcumin solutions since, without US, too high doses could have a cytotoxic effect. Therefore, a key step in our strategy was to use suitably low concentrations, showing a cytotoxic effect mainly in combination with ultrasounds. We also established the range of appropriate parameters of ultrasonic waves (power, intensity, duration of action), which showed no cytotoxic effects alone but significantly reduced cell viability when combined with curcumin. The cytotoxic effect of the selected intensity value (3 MHz) in our study was significantly lower than that described in the literature [8,12,64,65,66]. Concomitantly, the lower values (1–2 MHz) resulted in too high cytotoxicity of US against SCC-25 and FaDu neck and head cancer cell lines.

MTT assay was used to study the cytotoxicity of compounds tested and their combinations with US in SCC-25 and FaDu cells. Ultrasound parameters were selected based on available literature data [8,12,61,64,65,66]. As shown in Figure 3 and Figure 5, the viability of SCC-25 and FaDu cells after US exposure was inhibited by ~20% and ~13%, respectively.

The number of SCC-25 cells treated with oleanolic (**1**) alone was decreased by 27% as compared to the control (Figure 3). However, the treatment with **1** and US was shown to reduce the viability of SCC-25 cells by 45% compared to untreated ones. Similarly, the cytotoxicity of oleanolic acid oxime (**2**) was lower than that of **2** combined with US. On the contrary, we observed only slightly reduced viability of the SCC-25 cell line treated with Oleanolic acid oxime methyl ester (**3**) alone as well as **3** used in combination with US (~24–30%). The cytotoxic activity of the curcumin (**4**) was similar to that of **3**. However, the number of SCC-25 cells exposed to **4** and the US was reduced significantly by ~50%, as compared to the control. The viability of the SCC-25 cell line treated with both curcumin pyrazole (**5**) alone, as well as the combination of **5** and US, was slightly inhibited (~25%) compared to untreated cells. However, the latter one was not statistically significant. Oleanoyl hydrogen succinate (**6**) was shown to decrease the cell viability of SCC-25 cells by ~10% as compared to the control. On the contrary, a hybrid of curcumin pyrazole and oleanoyl hydrogen succinate (**7**) alone has no statistically significant effect on cell proliferation. However, the exposure to **6** combined with US, as well as the combination of **7** and US, reduced the number of viable cells by ~50% compared to the untreated control.

Additionally, the tendency of the sonotherapeutic effects of the tested compounds in relation to SCC-25 cell lines is presented in Figure 4.

As shown in Figure 5, the viability of FaDu cells was reduced by ~29% after treatment with the oleanolic acid (**1**). The US exposure in the presence of **1** was shown to decrease the number of viable cells by ~46% as compared to the control. The derivatives of OA, compounds **2** and **3,** were shown to similarly inhibit the viability of the FaDu cell line (~20%). The combination of US with **2** and **3** reduced the number of viable cells by ~34% and ~29%, respectively, as compared to the control. The cytotoxic activity of the curcumin was only slightly lower than that of **4** and UV. However, statistical significance was observed as compared to untreated cells. The curcumin pyrazole (**5**) alone reduced the viability of FaDu cells by ~34%, while the exposure to compound **5** combined with US inhibited the number of viable cells by 44% as compared to control. The succinyl derivative of OA (**6**) and the hybrid pyrazole curcumin and oleanoyl hydrogen succinate (**7**) were shown to slightly decrease cell viability (~10%) compared to the control; however, the combination of US with **6** and **7** inhibited the cell viability by ~60% and ~80%, respectively.

The observed difference between cell survival after the action of the tested compounds in monotherapy and cell viability after treatment in combination with ultrasounds suggests that separate mechanisms might be responsible for the cytotoxic effect. The results of the MTT test suggest that both the action of compounds **1**-**7** in monotherapy and the action of their combinations with ultrasounds might bring different effects in epithelial cell lines derived from different tumors. Comparing the results obtained in the SCC-25 tongue cancer cells and the FaDu throat ones, we cannot clearly state which cell line is more sensitive to the SDT since we showed a marked decrease in cell viability after treatment with compounds **1**-**7** alone and their combinations with US in both cell lines.

Taking into account the chemical nature of the tested compounds, curcumin (**4**) showed a better cytotoxic effect on SCC-25 cells than oleanolic acid (**1**), both in monotherapy and in combination with US. In the FaDu cell line, the effect of both compounds was comparable in monotherapy, while oleanolic acid (**1**) was found to exert significantly stronger cytotoxicity after exposure to US. In the group of oleanolic acid derivatives, oleanolic acid (**1**) showed the most potent monotherapy effect in FaDu cancer cells, while oleanoyl hydrogen succinate (**6**) indicated the higher cytotoxic effect in SDT therapy. However, when comparing curcumin and its pyrazole derivative, the latter revealed a stronger cytotoxic effect, both individually and in combination with US. In the group of triterpene compounds in the SCC-25 cell line, the highest cytotoxic activity in monotherapy was shown by oleanolic acid oxime (**2**), while after exposure to ultrasounds, the strongest effect was exerted by compound **6**. Comparing compounds **4** and **5**, curcumin (**4**) showed higher cytotoxicity against SCC-25 cells, both in monotherapy and in SDT. Compound **7**, a hybrid of curcumin pyrazole and oleanoyl hydrogen succinate, was found to be the most effective sonosensitizer in both tested cell lines (Figure 6).

## 4. Conclusions

Literature data and results of experimental work indicate that natural compounds, including curcumin and oleanolic acid derivatives, are highly potent substances that provide a unique and safe base for the development of new drugs.

In summary, different derivatives of oleanolic acid and curcumin were synthesized, and their sonosensitizing activities against SCC-25 and FaDu cells were evaluated. In the present study, we showed for the first time the cytotoxic effects of curcumin, oleanolic acid, and their derivatives combined with US in both human *squamous cell carcinoma* cell lines. The study of the combined effects of these two agents was carried out in parallel with the study of the effects of the organic compound and ultrasound alone.

Among the triterpene derivatives, the following were tested: oleanolic acid (**1**), its oxime (**2**), its methyl ester oxime (**3**), and oleanoyl hydrogen succinate (**6**). In parallel, curcumin (**4**) and its heterocyclic pyrazole (**5**) were evaluated. A complex hybrid derivative (**7**) combining the above-mentioned structures of triterpene and curcumin in one molecule was also obtained and investigated.

Based on the available literature data and above all our results, we suggest that oleanolic acid, curcumin, and their derivatives, especially the triterpene–curcumin type hybrid (**7**), exhibit a high sonosensitizing potential and have a chance to be used in SDT therapy supporting the treatment of head and neck cancers.

## Data Availability

Not applicable.

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
