# Peer review of "Curcumin–Triterpene Type Hybrid as Effective Sonosensitizers for Sonodynamic Therapy in Oral Squamous Cell Carcinoma"

_pharmaceutics, 2023, doi:10.3390/pharmaceutics15072008_

Round 1
Reviewer 1 Report
The submitted manuscript pharmaceutics-2498213 entitled as “Curcumin-triterpene type hybrid as effective sonosensitizers for sonodynamic therapy in oral squamous cell carcinoma” by K. Sowa-Kasprzak, M. Józkowiak, D. Olender, A. Pawełczyk, H. Piotrowska-Kempisty and Lucjusz Zaprutko focused on synthesis of some new sonosensitizers based on different derivatives of oleanolic acid and curcumin as well as in vivo evaluation of the sonodynamic effect of them into tongue cancer SCC-25 and hypopharyngeal FaDu cell lines.
The main idea of the manuscript is clear as well as design of the present study. I have not of any questions for authors and I would like recommend accept of this manuscript in present form.
Author Response
We would like to thank you for your opinion and kind comment.
Reviewer 2 Report
The manuscript deals with the in vitro evaluation of the sonodynamic effect of natural compounds. In particular, curcumin, oleanolic acid, and various chemically modified derivatives were applied as sonosensitizers in Sonodynamic Therapy to cancerous SCC-25 and FaDu cell lines. According to the study, the highest sonotherapeutic activity, compared to the individual components, was demonstrated by a structural hybrid complex molecule containing both curcumin pyrazole and oleanoyl hydrogen succinate moieties.
In general, the paper is well-written and offers interesting ideas. However, it appears a bit superficial because the authors didn’t make any hypothesis about how the proposed sonosensitizers work. I mean, for example, is the cell death mechanism apoptotic? Or is it necrotic? This information could help the comprehension of the phenomena.
Minor concerns are listed below:
-Page 4: I don't know if the different line spacing between the two sections (Materials and Methods and Results and Discussion) is wanted or not. However, it should be better to standardize it.
-Line 286: What figure 1 and 2 do the authors refer to? Here Figure 1 shows the molecular structures of two sonosensitizers, and Figure 2 shows the scheme of functioning of US.
Please check the manuscript for minor English mistakes.
Author Response
First, we would like to thank you for your opinion and kind comment.
The manuscript deals with the in vitro evaluation of the sonodynamic effect of natural compounds. In particular, curcumin, oleanolic acid, and various chemically modified derivatives were applied as sonosensitizers in Sonodynamic Therapy to cancerous SCC-25 and FaDu cell lines. According to the study, the highest sonotherapeutic activity, compared to the individual components, was demonstrated by a structural hybrid complex molecule containing both curcumin pyrazole and oleanoyl hydrogen succinate moieties.
In general, the paper is well-written and offers interesting ideas. However, it appears a bit superficial because the authors didn’t make any hypothesis about how the proposed sonosensitizers work. I mean, for example, is the cell death mechanism apoptotic? Or is it necrotic? This information could help the comprehension of the phenomena.
The cell viability tests performed within this article framework were intended to be screening tests. Unfortunately, based only on the preliminary studies, we cannot suggest any hypothesis regarding their mechanisms of cytotoxic activity. However, the thorough understanding of the mechanisms that drive their cytotoxicity in the SCC-25 and FaDu cell lines may become an interesting subject of further research.
Minor concerns are listed below:
-Page 4: I don't know if the different line spacing between the two sections (Materials and Methods and Results and Discussion) is wanted or not. However, it should be better to standardize it.
It has been corrected.
-Line 286: What figure 1 and 2 do the authors refer to? Here Figure 1 shows the molecular structures of two sonosensitizers, and Figure 2 shows the scheme of functioning of US.
We agree this is a mistake. It has been changed to: "in Figures 3 and 5".
Comments on the Quality of English Language
Please check the manuscript for minor English mistakes.
The manuscript was checked for proper English language.
Reviewer 3 Report
The manuscript "Curcumin-triterpene type hybrid as effective sonosensitizers for sonodynamic therapy in oral squamous cell carcinoma" shows promising results from the combination of compounds with sonodynamic therapy. However, authors must answer the following questions:
· The abstract needs to be improved. They must incorporate the results, showing the best values obtained.
· For feasibility studies, because non-tumor cell lines were not incorporated, to determine if the treatment combination can exert any negative effect on healthy cells.
· Because cytotoxicity studies (LDH assay) were not performed, since the conditions of the sonodynamic therapy (resonance frequency of 3MHz and ultrasonic intensity of 0.8 W/cm2) - can influence cell membranes.
· Because the compounds curcumin and oleanolic acid were chosen, and not others such as gases (H2, CO2, and N2)
· Because a positive control (antitumor drug) was not added to the viability tests, in order to compare the values obtained.
Authors should check the language. There are parts in American English and British English.
Author Response
First, we would like to thank you for your opinion and kind comment.
The manuscript "Curcumin-triterpene type hybrid as effective sonosensitizers for sonodynamic therapy in oral squamous cell carcinoma" shows promising results from the combination of compounds with sonodynamic therapy. However, authors must answer the following questions:
- The abstract needs to be improved. They must incorporate the results, showing the best values obtained.
It has been improved. Changes were highlighted in the text.
- For feasibility studies, because non-tumor cell lines were not incorporated, to determine if the treatment combination can exert any negative effect on healthy cells.
The cytotoxicity assays performed on SCC-25 and FaDu cell lines were intended to be screening tests. Further studies, regarding the mechanisms of activity of the most potent ones, will include the non-tumor cell line to evaluate the effects on healthy cells.
- Because cytotoxicity studies (LDH assay) were not performed, since the conditions of the sonodynamic therapy (resonance frequency of 3MHz and ultrasonic intensity of 0.8 W/cm2) - can influence cell membranes.
Instead of LDH assay, which can be applied to measure the number of dead cells, we performed the MTT test. Due to performing the MTT test, we were able to measure the number of viable cells in culture and thus determine the cytotoxicity of tested compounds.
- Because the compounds curcumin and oleanolic acid were chosen, and not others such as gases (H2, CO2, and N2)
The experiments were carried out in normal atmospheric conditions without the addition of gases changing the atmosphere, based on previous literature reports and own experience.
- Because a positive control (antitumor drug) was not added to the viability tests, in order to compare the values obtained.
We would like to thank the Reviewer for the suggestion. The comparison with antitumor drug, as a positive control, will enrich the studies regarding specific mechanisms of action of chosen compounds. However, in the present study, we only intended to assess their cytotoxicity.
Comments on the Quality of English Language
Authors should check the language. There are parts in American English and British English.
It has been corrected.